# Bio-Based White Eggshell as a Value-Added Filler in Poly(Lactic Acid) Composites

**Duncan Cree** [1,*] and **Majid Soleimani** [2]

1 Department of Mechanical Engineering, University of Saskatchewan, Saskatoon, SK S7N 5A9, Canada
2 Department of Chemical and Biological Engineering, University of Saskatchewan, Saskatoon, SK S7N 5A9, Canada; mas233@mail.usask.ca
* Correspondence: duncan.cree@usask.ca; Tel.: +1-306-966-3244

**Abstract:** Based on its positive environmental impact, poly(lactic acid) (PLA) has been a gradual substitute for synthetic plastics used in diverse applications. The use of industrial limestone (ILS) as a filler in polymers can have advantages of changing the properties of pure polymers. Waste eggshells (WE) can be seen as an alternative filler to ILS as they are also a source of calcium carbonate. To assess the feasibility of both filler types and sizes, PLA composites were manufactured by injection molding with filler contents of 5, 10, and 20 wt.%. Tensile, flexural, and impact mechanical properties were evaluated in addition to water absorption. One-way analysis of variance (ANOVA) was performed to determine whether statistically significant differences among the measured mechanical properties existed. Scanning electron microscopy (SEM) was used to view the morphology of the fillers and fractured surfaces. The composite tensile strengths and flexural strengths performed the best when filler loadings were 5 wt.% and 10 wt.%, respectively, for both filler types. The tensile and flexural modulus both increased with filler loadings. The impact strength for the composites was obtained at a threshold level of 5 wt.% filler loadings for both filler types and slightly better for smaller particles sizes. ANOVA identified statistically significant differences for the mean mechanical property values evaluated. SEM showed the fractured surfaces of the PLA composites were different from the pure PLA indicating some transformation occurred to the matrix. The weight gains due to water absorption were observed to increase with increase in content of both filler types while the smaller particles had slightly higher water weight gains. Although the composites containing ILS fillers had somewhat enhanced mechanical properties over the WE-filled composites, the end application will dictate which filler type to use in PLA.

**Keywords:** mechanical properties; industrial limestone; white eggshells; composite; injection molding; PLA

## 1. Introduction

Modern society consumes plastics that are not renewable at a rate that is not sustainable. Thermoplastics are derived from petroleum and can be recycled where recycling industries exist, but many synthetic plastics around the world end up in landfills and oceans as pollution. An alternative eco-friendly, sustainable material is poly(lactic acid) (PLA), which is a bio-degradable polymer derived from corn, wheat, or rice [1]. PLA was initially discovered in 1845 by French chemist Theophile Jules Pelouze who synthesized PLA by the poly-condensation of lactic acid. Later, the process was improved by chemist Wallace Hume Carothers and patented by DuPont in 1954, but at that time PLA was expensive. It was not until 1989 with chemist Patrick R. Gruber, founder of Cargill Dow LLC, that the commercial use for the material was recognized due to the discovery of a lower-cost manufacturing method by lactide ring opening polymerization [2]. PLA is considered carbon neutral since any carbon dioxide ($CO_2$) produced as a by-product from the process or during biodegradation is consumed by regrowth of subsequent biomass.

After its useful life, PLA can be biodegraded without producing toxic by-products through composting given the right conditions of temperature, moisture, soil pH, bacteria and other parameters as outlined by Araújo et al. [3]. Another approach to recycle PLA is to use chemical hydrolysis, which uses water and elevated temperature to break down PLA into monomers of high-grade lactic acid. These monomers can then be used in the production of pure PLA, which will result in the same properites as the conventional route. This process avoids a more expensive glucose fermentation, a traditional process often used for obtaining the pure/raw lactic acid [4].

Industrial limestone (ILS), a carbonate sedimentary rock, is made of calcium carbonate ($CaCO_3$) with calcite as the major mineral. They are extracted from quarries and used widely in conventional thermoplastic polymer composites not only because they are low-cost fillers, but they can improve stiffness and toughness compared to pure polymers [5]. Calcium carbonate was added to high-density polyethylene (HDPE) in amounts of 5, 10, and 20%, where the tensile modulus was observed to increase for all particle contents, while the impact toughness improved for all samples within the temperature range of $-40$ to $+70$ °C compared to pure HDPE. The study determined there was no indication of particle aggregation as the particles were uniformly distributed within the HDPE matrix. Interestingly, chemical coatings or coupling agents to improve filler distribution, aggegation/agglomeration, and compatibility with the polymer matrix were not applied to the calcium carbonate particles [6]. In similar ways, other studies have reviewed thermal, mechanical, and physical properties of limestone fillers in HDPE [7–9], in low-density polyethylene (LDPE) [7,10–12], in polypropylene (PP) [13,14], and in PLA [15].

In order to maintain the focus of sustainable polymer products, limestone could potentially be replaced in full or in part by a bio-based calcium carbonate material source such as powdered eggshells. Home and restaurant consumption of table eggs would not be a viable source of calcium carbonate. However, egg breaking plants such as Burnbrae Farms[TM], Egg Solutions EPIC Inc., and Rembrandt Foods[®] in North America use an industrial process to remove liquid from eggs in substantial quantities. The current practice discards eggshells to landfills at a disposal fee to the company. One dried white chicken eggshell weight is approximately 6.6 g and contains 96.9% $CaCO_3$, a sustainable source of calcium carbonate with a remaining 1–3% organic matter [16]. Interestingly, eggshells have a higher mineral purity than limestone, since mined limestone can contain inherent impurities based on the geological formation. The $CaCO_3$ purity of industrial limestone can range from 90–99% [17]. In 2021, Canada's annual egg production was 848.2 million dozen [18] or 10.2 billion eggs. Breaking plants consume about 30% of the total egg production, which would result in 3.1 billion eggs. Therefore, the annual production of limestone originating from eggshells is about 20,500 tons. While this is not a substantial amount, the goal is to recuperate a waste material and save breaking plants from disposal costs to landfills. Recently companies such as Eggshell Resources Inc. (Thornhill, ON, Canada) and Just Egg-Chilled Foods Ltd. (Leicester, UK) showed interest to recuperate and purify eggshell waste for its $CaCO_3$ content and organic membrane ingredients for use in pharmaceutical and food packaging industries. A number of different uses of eggshells have been recently documented. For example, patents have been obtained for use as a fertilizer, powder for chalk, absorbent for removal of heavy metals from soil and water, and facial cleansers [19]. However, in the literature, there have been limited studies on bio-based limestone, compared to conventional limestone as fillers in synthetic polymers. For instance, eggshells [20–23] and clam seashell powder [24] were added as filler materials in PP matrices. Similarly, eggshells have been incorporated into LDPE [25,26], HDPE [27], and poly(vinyl chloride) [28].

A limited number of studies have investigated biodegradable polymers containing eggshell particles as fillers thereby producing a fully bio-composite material. A study was conducted on producing PLA films containing 1, 2, 3, 4, and 5 wt.% eggshell particles having a size of 25 μm. The addition of filler improved the tensile strengths by 5.4, 25.6, 51.8, 82.1, and 33.9%, respectively, while the modulus increased by 1.7, 24.3, 53.4, 70.5, and 20.6%,

respectively. It was shown agglomeration of particles was significant for filler contents of 5 wt.% [29]. In another study, eggshell powders were added by an extrusion process to Bioplast®, a bio-based thermoplastic in amounts ranging from 1–3 wt.%. The eggshells were crushed, ground, and further decreased in size using a sono-chemical method to produce nano-sized particles less than 10 μm. The flexure properties increased for all filler contents compared to the pure Bioplast. The highest improvement was observed with the addition of 2 wt.% filler where the flexural strength and modulus increased by 35.3% and 30.5%, respectively [30]. Extrusion of PLA/eggshell fillers into filament for 3D printing has recently been investigated [31,32].

In this study, efforts have been carried out to investigate the mechanical and physical properties of thermoplastic PLA as the matrix material to compare the additions of two filler materials: industrial limestone (ILS) and white eggshell (WE) powders. The composite materials were manufactured using a twin-screw extruder followed by injection molding. One-way ANOVA was performed on the mechanical properties to identify statistically significant differences in the composites. Finally, water absorption, composite weight loss, and distilled waters were analyzed for calcium content and pH changes after 35 days. The innovation of this work lies in the unique combination for the selection of biopolymer PLA, filler particle size, filler contents, and manufacturing process.

## 2. Materials and Methods

### 2.1. Materials

A commercial extrusion-grade injection molding Ingeo™ biopolymer PLA 4043D in pellet form was purchased from NatureWork® LLC, Minnetonka, MN, USA as the matrix. The term "pure PLA" was the 4043D PLA grade and used to differentiate between the different PLA composites in this study. The filler materials were industrial limestone (ILS) from Imasco Minerals Inc. (Surrey, BC, Canada) and white chicken eggshell (WE) obtained from Burnbrae Farms™ (Brockville, ON, Canada).

### 2.2. Eggshell Processing

The as-received white eggshells arrived semi-crushed as shown in Figure 1a. The eggshells were then coarse crushed, agitated, and washed numerous times to remove the organic membranes as shown in Figure 1b, followed by drying. This procedure was thoroughly described in a related study where it was determined that although this method removed most of the organic membrane, about 1.3% remained [33]. A ball mill was used in a final step to refine the particle size as shown in Figure 1c. Finally, both limestone and white eggshell powders were sieved using 63 μm and 32 μm mesh size sieves.

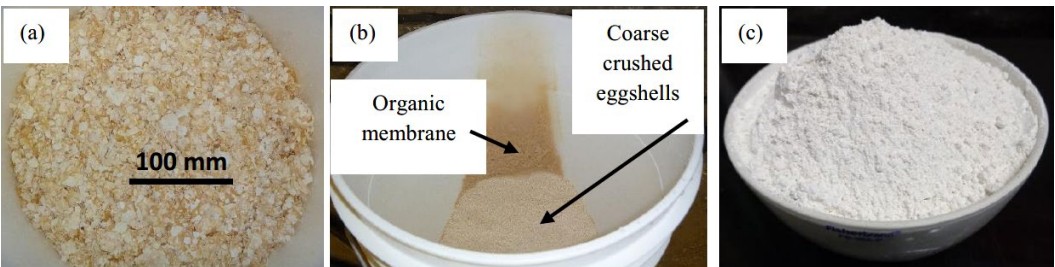

**Figure 1.** (**a**) As-received semi-crushed eggshells, (**b**) washing/rinsing step, and (**c**) refined eggshell powder.

### 2.3. Preparation of Composites

The PLA composites were initially blended using a mixer, then processed by melt-blending in a twin-screw extruder (SHJ-35, Nanjing Yougteng Chemical Equipment Co. Ltd., Nanjing, China) at a temperature of 175 °C. After cooling, the extrudates were pelletized. Prior to injection molding, all ingredients were dried in an oven at 80 °C for 4 h to remove moisture. The pellets were used in an injection molding machine (Shen Zhou (SZ) 2000,

Zhangjiagang Shen Zhou Machinery Co., Shenzhen, China) using a temperature profile of 175, 180, 185, and 190 °C from feed zone to die. A typical water absorption (flat plate) and tensile specimen (dog-bone) is shown in Figure 2. The flat central portion of the tensile samples were also used for flexural and impact tests. The PLA composites were blended into various compositions using 5, 10, and 20 wt.% powders as shown in Table 1 for both particle sizes.

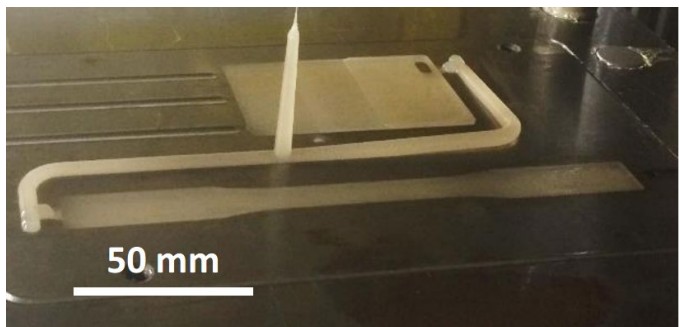

**Figure 2.** Typical injection molded water absorption (flat plate) and tensile (dog-bone) specimens.

**Table 1.** Composition and description of PLA composites containing industrial limestone (ILS) or white eggshell (WE) $CaCO_3$ fillers.

| Material | ILS or WE (wt.%) | PLA (wt.%) |
|----------|------------------|------------|
| Pure PLA | 0 | 100 |
| LS-5 | 5 | 95 |
| LS-10 | 10 | 90 |
| LS-20 | 20 | 80 |
| WE-5 | 5 | 95 |
| WE-10 | 10 | 90 |
| WE-20 | 20 | 80 |

*2.4. Composite Characterization*

2.4.1. Mechanical Testing

Tensile tests were performed with ASTM D638-14 using an Instron universal testing machine (model 1137) with a load cell of 10 kN and a strain rate of 5 mm/min. Specimens measured 200 mm × 12.74 mm × 3.25 mm (length × width × thickness). Flexural three-point bend tests were carried out according to ASTM D790-17 using the Instron universal testing machine (model 1137) with a load cell of 250 N and a rate of 1.3 mm/min. Specimens measuring 65 mm × 12.74 mm × 3.25 mm were obtained from the center of tensile samples (length × width × thickness). Impact, un-notched tests were conducted with guidance of ASTM D6110-18 using an Instron 450 MPX impact testing machine. Un-notched samples were obtained from the middle portion of tensile specimens with dimensions of 55–56 mm × 12.74 mm × 3.25 mm (length × width × thickness). Before testing, all samples for mechanical testing were preconditioned at 22 ± 3 °C and a relative humidity of 49 ± 3% for 24 h. For statistical purposes, tensile and flexural test results were an average of five samples each, while the impact samples were an average of ten samples.

2.4.2. Morphological Analysis

The morphology of the filler particles and fractured surfaces (derived from remnants after testing) of the composites were examined by scanning electron microscopy (SEM). SEM was carried out using a JEOL JSM-6010 LV (Tokyo, Japan) with an operating voltage of 10–20 kV. The samples were mounted on small stubs and coated with a thin layer of gold using a sputter coater to improve the conductivity.

### 2.4.3. Statistical Analysis

The tensile, flexural, and impact strengths as well as the tensile and flexural modulus were each analyzed separately using an F-test in the form of a one-way (or one factor) analysis of variance (ANOVA) at a 5% level of significance (95% confidence level) to determine if the addition of the calcium carbonate filler materials had a statistical effect on the mechanical properties of the different composite groups. The only variable or factor changing was the filler loading. Each mechanical property was evaluated as one group which consisted of 13 groups where Group 1 was for pure PLA samples, Groups 2–7 for composites with 63 µm fillers, and Groups 8–13 for composites with 32 µm fillers. The analysis was performed using the Analysis ToolPak add-in feature in Microsoft Excel 365 (Microsoft Corp., Redmond, WA, USA). If the F-critical value was less than the F-value, the results suggest significant differences of means exist between the composite groups (rejection of null hypothesis). If the F-value is close to zero or close to the F-critical value, it suggests there were no differences in the mean values obtained.

### 2.4.4. Water Absorption

Rectangular test samples were cut using a band saw to dimensions of 57.4 mm × 36.4 mm × 2.7 mm (length × width × thickness). The cut edges were then made smooth using a 120-grit silicon carbide paper. Prior to testing, the specimens were first conditioned by drying them at 50 °C for 24 h, cooled in a desiccator, and weighted to the nearest 0.001 g. Water absorption was performed following ASTM D570–18 where three samples were totally submerged in distilled water at room temperature for 35 days. At 35 days, there was no significant change in weight which suggested the samples were at equilibrium. Before weighing the samples, the surface was blotted gently with a paper towel to remove surface water. The percent increase in weight gains due to water absorption for both composites contain 63 µm and 32 µm fillers were calculated using Equation (1), while the weight loss after 35 days of water exposure was determined for composites containing 32 µm fillers using Equation (2):

$$\text{Water absorption } (\%) = \frac{(W_w - W_d)}{W_d} \times 100\% \tag{1}$$

$$\text{Weight loss } (\%) = \frac{(W_t - W_d)}{W_d} \times 100\% \tag{2}$$

where, $W_w$, $W_t$, and $W_d$ are the wet weight after exposure to water, the dry weight after exposure to water, and the initial dry weight before exposure to water, respectively. To evaluate the calcium content due to the possibility of calcium carbonate leaving the PLA composite materials, water samples from the respective beakers were analyzed after 35 days using an atomic absorption spectrophotometer (model 240 Series AA, Agilent Technologies, Santa Clara, CA, USA) to determine the calcium ion migration into the water.

### 2.4.5. pH Change of PLA Composites

The pH change was measured in beakers containing pure PLA and PLA composites with 32 µm calcium carbonate particles, which were immersed for a total of 35 days using a pH meter (Orion Star$^{\text{TM}}$ A111 pH Benchtop Meter; Thermo Scientific Ward Hill, MA, USA). The distilled water was not refreshed throughout the 35-day soaking period. To prevent water evaporation, the beakers were covered with silicone caps sealing the contents of the beaker. The temperature of the water on Day 35 was 21.1 °C. The average of three pH measurements for each composition was reported.

## 3. Results and Discussion

### 3.1. Mechanical Properties

3.1.1. Tensile Property

The tensile strength and tensile modulus results are shown in Figure 3 for pure PLA and PLA composites containing industrial limestone (ILS) and white eggshell (WE) as fillers in amounts of 5, 10, and 20 wt.% for two particle sizes 63 μm and 32 μm. The manufacturer data sheet values for 4043D PLA were included for comparison. The 4043D PLA and the pure PLA (e.g., the 4043D that was injection molded in this study) had comparable tensile strengths of 53.0 MPa and 50.9 MPa, respectively, with the same tensile modulus of 3.6 GPa.

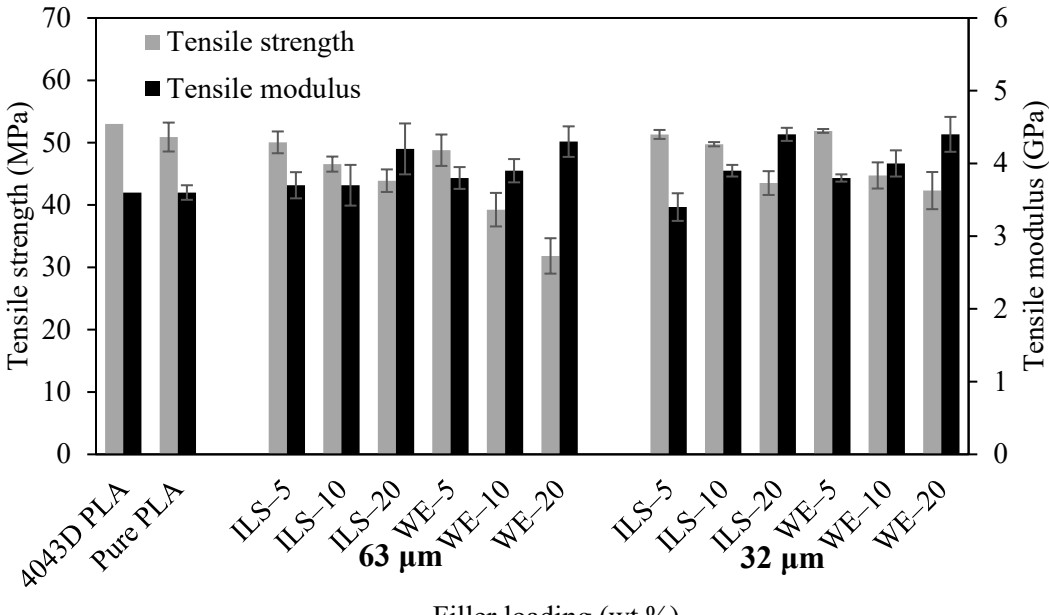

**Figure 3.** Influence of filler loading on tensile strength and tensile modulus of PLA composites.

Compared to pure PLA, the tensile strength of the composites generally decreased with increase in filler concentration. For composites containing 63 μm ILS and WE fillers in loadings of 5, 10, and 20 wt.%, the tensile strengths reduced by 1%, 9%, and 14% and 4%, 23%, and 38%, respectively, while the 32 μm ILS and WE fillers improved slightly by 1% and 2%, respectively at 5 wt.% loading but reduced by 2% and 15% for ILS and 12% and 17% for WE for 10 and 20 wt.% loadings, respectively. The tensile strengths of the ILS composites were marginally superior to the WE composites where a 5 wt.% filler loading reduced the least for both filler types. When adding filler particles to a polymer matrix, the tensile strengths of the composites tend to drop. As the calcium carbonate filler contents increase, there are more particles per unit volume, which reduces the load-bearing cross-section of the PLA matrix material, thus reducing the overall composite tensile strength [34]. Tensile strength is affected by aggregation/agglomeration of microparticles. The reduction in tensile strength may be attributed to random particles joining together by weak van der Waal forces to form larger (weaker) aggregates during the manufacturing process. In a related study, although efforts were made to blend calcium carbonate filler particulates into a PLA polymer, the dispersion of micron-sized fillers in the polymer matrices may not be perfect as was also reported by Tjong et al. [35].

The tensile modulus increased with filler loadings. There was a gradual improvement in tensile modulus for composites containing 63 μm fillers in amounts of 5, 10, and 20 wt.% loadings. The composites containing ILS increased by 3%, 3%, and 17%, respectively, while those containing WE fillers improved by 6%, 8%, and 19%, respectively. The composites containing 32 μm ILS fillers decreased by 6% at 5 wt.% loading but increased by 8% and 22% at 10 and 20 wt.% loadings, respectively. A study on the addition of eggshell

fillers in polypropylene found well-dispersed fillers increased the tensile modulus of the composites [22]. It is reasonable that during the injection molding process, the ILS fillers at 5 wt.% were not well distributed within the PLA. The composites with 32 μm WE fillers in loadings of 5, 10, and 20 wt.% increased in tensile modulus by 6%, 11%, and 22%, respectively. For both particle filler size and filler types, the highest tensile modulus was reached at 20 wt.% loadings. Both ILS and WE fillers are based on calcium carbonate, which is a stiff material compared to the less rigid PLA polymer. The tensile modulus is sensitive to polymer chain-filler interactions. The stiff fillers distributed between the polymer chains benefits the tensile modulus by restricting the polymer chain mobility [8]. For example, the tensile modulus of mineral micron-sized calcium carbonate has been reported to be in the range of 25–50 GPa [36], in contrast to 3.6 GPa for PLA. Both tensile strength and tensile modulus results were controlled more by the filler loading rather than the particle size.

### 3.1.2. Flexural Property

The flexural strength and flexural modulus results are shown in Figure 4 for PLA composites containing 63 μm and 32 μm sized ILS and WE fillers in concentrations of 5, 10, and 20 wt.%. To compare against the manufacturer's data sheet values, the 4043D flexural properties were added to Figure 4. The 4043D PLA and the pure PLA (e.g., the 4043D that was injection molded in this study) had flexural strengths of 83.0 MPa and 78.9 MPa, respectively, while the tensile modulus was 3.8 GPa and 2.9 GPa, respectively.

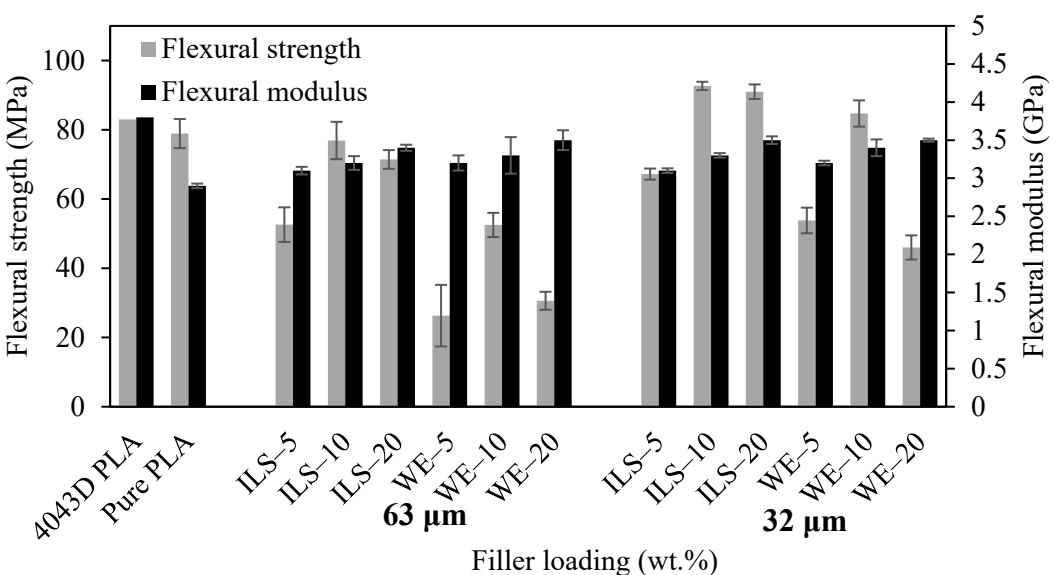

**Figure 4.** Influence of filler loading on flexural strength and flexural modulus of PLA composites.

Depending on the choice of filler and of polymer matrix, there is an optimal filler content which leads to a well dispersion of particles within the matrix material. For the current study, the flexural strengths were observed to decrease with filler loading but performed the best at 10 wt.% filler concentrations. The industrial limestone fillers slightly outperformed the eggshell filler composites in their flexural properties. For instance, compared to pure PLA, composites containing 63 μm fillers in loadings of 5, 10, and 20 wt.% reduced in flexural strength by 33%, 3%, and 10%, respectively, for ILS-filled composites. In a similar way, flexural strengths decreased by 67%, 34%, and 61%, respectively, for WE-filled composites. However, composites with additions of 32 μm ILS fillers initially decreased by 15% at 5 wt.%, but improved by 18% and 15%, at 10 and 20 wt.%, respectively. The composites consisting of 32 μm WE fillers tended to reduce the flexural strength by 32% at 5 wt.% content, but increased by 7% at 10 wt.%, followed by a drop of 42% at 20 wt.%. Larger particles have a low surface energy, which leads to better dispersion and reduced agglomeration, but a smaller surface area acts to reduce contact/interaction with the PLA

matrix. Therefore, at the interface, less stress is transferred from the matrix to the filler, which reduces the mechanical properties. The results show there is an optimum particle size and filler loading combination, which enhances the flexural strength. As filler loadings increase, the risk of agglomeration and resulting stress concentrations is expected, which would reduce the flexural strengths.

Similar to the tensile modulus, the flexural modulus increased with filler concentration for both particle sizes and types of filler. Composites with filler loadings of 5, 10, and 20 wt.%, improved in flexural modulus by 7%, 10%, and 17%, respectively, for 63 μm ILS particle composites, while 63 μm WE filled composites increased in flexural modulus by 10%, 14% and 21%, respectively. When smaller particles sizes of 32 μm were added as fillers, the flexural modulus of composites with ILS fillers gained by 7%, 14%, and 21%, respectively. Similarly, composites with WE fillers increased in flexural modulus by 10%, 17%, and 20%, respectively. Adding more rigid fillers to the PLA tended to improve the stiffness of the composite materials. In general, it was observed that smaller particle sizes had better flexural strength and stiffness properties possibly due to their reduced size. Small filler particles characterized as having a high surface energy reduces dispersion and increases tendency to agglomerate, but a large surface area allows for increased interfacial contact/interaction between the filler and PLA matrix. The interface area supports the transfer of stress from the matrix to the filler particles leading to improved mechanical properties.

### 3.1.3. Impact Property

Figure 5 illustrates the impact strength with filler loading for PLA composites containing ILS and WE fillers for two particles sizes of 63 μm and 32 μm. The pure PLA (e.g., the 4043D that was injection molded in this study) samples had an impact strength of 17.3 kJ·m$^{-2}$, which is in line with the literature impact strengths (un-notched Charpy) for PLA being between 19 kJ·m$^{-2}$ [37] and 25 kJ·m$^{-2}$ [15]. PLA composites in loadings of 5, 10, and 20 wt.% reduced in impact strength for composites containing 63 μm ILS fillers by 24%, 36%, and 41%, respectively, while 63 μm WE fillers were lowered by 49%, 53%, and 57%, respectively. Similarly, composites with 32 μm ILS fillers decreased in impact strength by 15%, 19%, and 35%, respectively, whereas 32 μm WE filled composites dropped by 43%, 49% and 53%, respectively. There were no major differences in the behavior of PLA composites since the impact strengths reduced for both calcium carbonate filler types as the filler loading increased. Very little to no toughening ability of the calcium carbonate particles above that of pure PLA was observed as was reported elsewhere for PLA/calcium carbonate unnotched composites containing 20 and 40 wt.% loadings with particle size of 1.89 μm. For example, there was an approximate 33% and 89% reduction, respectively, in impact strength compared to the pure PLA [15]. In the current study, the maximum impact strength was found at a threshold level of 5 wt.% filler loading. Overall, the composites containing 32 μm (smaller) particle size fillers decreased slightly less in impact strength than composites incorporating larger 63 μm particles. It has been reported that the impact property of polymers containing small, low aspect ratio particles are favored over large, high aspect ratio particles as they can act as flaws creating stress concentrations near their edges [38].

### 3.1.4. Morphological Analysis

The composites containing ≤ 32 μm particles performed slightly better than those containing ≤ 63 μm. Therefore, SEM images (Figures 6–9) were reported for composites containing industrial limestone and white eggshell of particle sizes ≤ 32 μm. Figure 6 shows the SEM of the industrial limestone (Figure 6a) and white eggshell (Figure 6b) particles sieved to sizes of ≤ 32 μm. In a previous study, a Malvern Mastersizer 2000 S (long bench) laser diffraction particle size analyzer was used to evaluate the ILS and ES particle sizes after sieving to 32 μm (No. 450) mesh. The results indicated particle sizes for ILS and WE were 25.1 ± 2.2 μm and 21.2 ± 2.0 μm, respectively [39]. Both filler particulates had a mixture

of jagged edges and irregular shapes. Also, both had a varying particle size distribution due to the crushing and grinding processes used. Interestingly, at higher magnifications, the eggshell particles illustrated an abundance of small pores on the material's surface as shown in Figure 6c, which was not observed in the industrial limestone. In the physiology of a chicken eggshell, these tiny pores function to allow oxygen to enter the shell and allow the exit of carbon dioxide ($CO_2$) for survival of the baby chick [40].

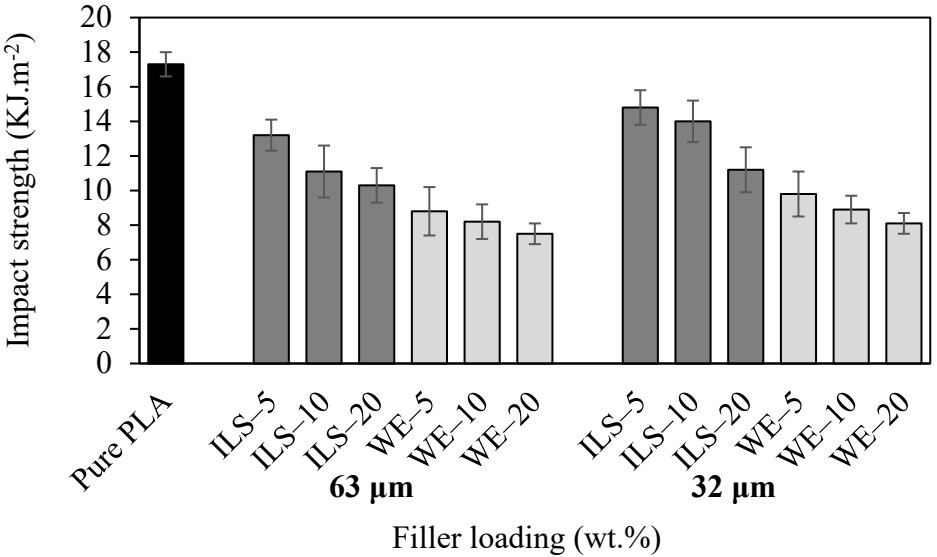

**Figure 5.** Influence of filler loading on impact strength of PLA composites.

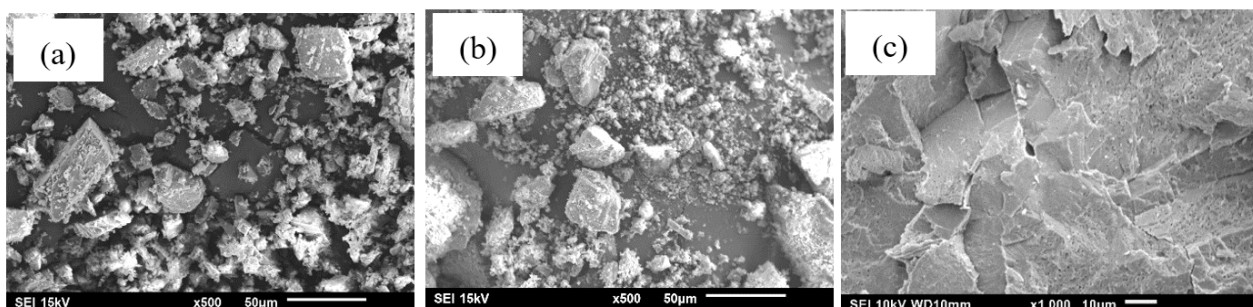

**Figure 6.** SEM images of particle surfaces for (**a**) industrial limestone, (**b**) white eggshells, and (**c**) magnified area of eggshell particle.

Figure 7 shows representative SEM images of the tensile fractured surfaces of pure PLA and PLA composites containing particles sizes ≤ 32 μm with minimum (5 wt.%) and maximum (20 wt.%) filler loadings. The pure PLA (Figure 7a) showed a rather smooth surface indicating brittle fracture, surrounded by white ridges representative of ductile fracture. With the addition of fillers, the fractured surface morphologies are different. From a visual inspection of the SEM fractures in Figure 7, red arrows identify filler particles and black arrows point out the voids. The voids indicate regions where filler particles were positioned within the matrix prior to fracture.

The flexural fractured cross-section features of PLA and PLA composites containing ≤ 32 μm particle sizes are shown in Figure 8. The fractured surface of pure PLA (Figure 8a) had relatively larger areas that were smooth and flat, which indicates the matrix is more brittle compared to the smaller regions of white ridges depicting limited ductility.

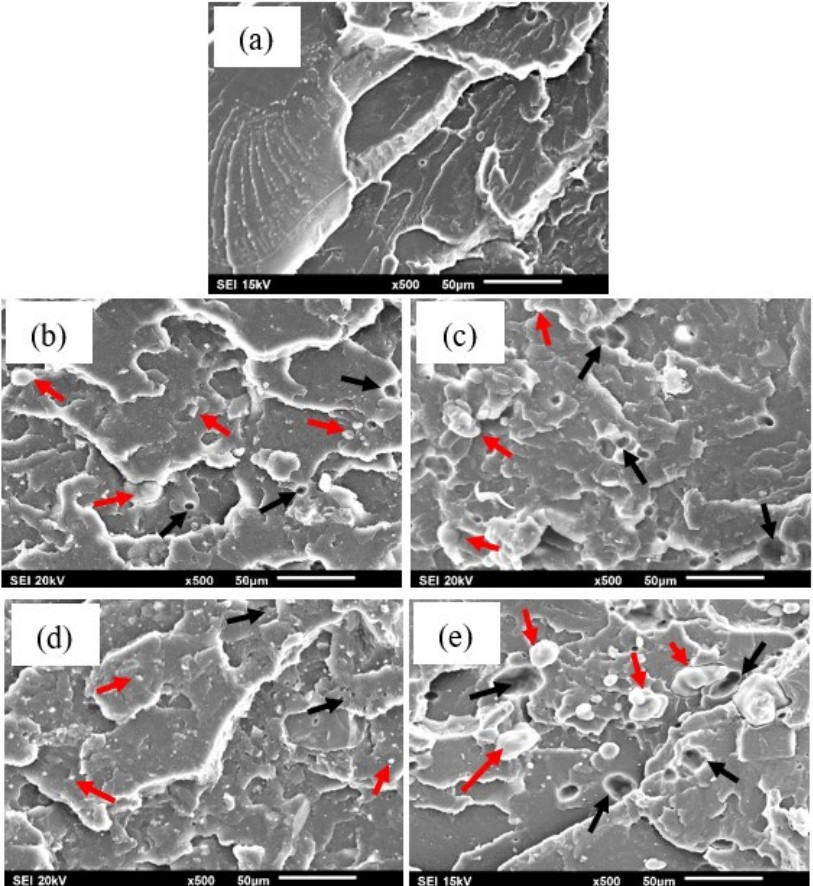

**Figure 7.** SEM images of the tensile fractured surfaces of PLA/CaCO$_3$ composites: (**a**) PLA, (**b**) PLA/ILS-5, (**c**) PLA/WE-5, (**d**) PLA/ILS-20, and (**e**) PLA/WE-20. Red arrows indicate particles and black arrows indicate voids.

The features of the fractured surface are modified with the addition of fillers. In both 5 wt.% (Figure 8b,c) and 20 wt.% (Figure 8d,e) micrographs, the particles appear to be de-bonded from the PLA matrix, which occurred during the deformation of the loading process. The softer PLA experienced deformation and flowed around the harder calcium carbonate particles. Compared to the tensile fractured SEM images, the flexural fractures (Figure 8b–e) illustrate a more obvious matrix–particle debonding, while the white ridges appear as coarse fibrils around the perimeter of the particles. The coarse fibrils are more elongated than the ridges observed in pure PLA, representing additional plastic deformation in the matrix. This observation is apparent on the fractured flexural samples compared to the tensile samples which may be a result of the bending action of the specimen during loading, rather than a longitudinal elongation during tensile loading.

Figure 9 shows the fractured surfaces of the impact test for PLA and PLA composites containing ≤ 32 μm sized particles. Pure PLA without filler highlights characteristics of brittle/ductile fractures in line with the fractured tensile and flexural specimens. Figure 9a shows smooth surfaces separated by river-like line patterns depicted by white ridges. By adding 5 wt.% calcium carbonate (Figure 9b,c), the river lines are still visible with slightly rougher surfaces due to the filler particles acting as obstacles for crack propagation. At low filler loadings (Figure 9b,c), adhesion between the particles and matrix is inferred as few particles are debonded.

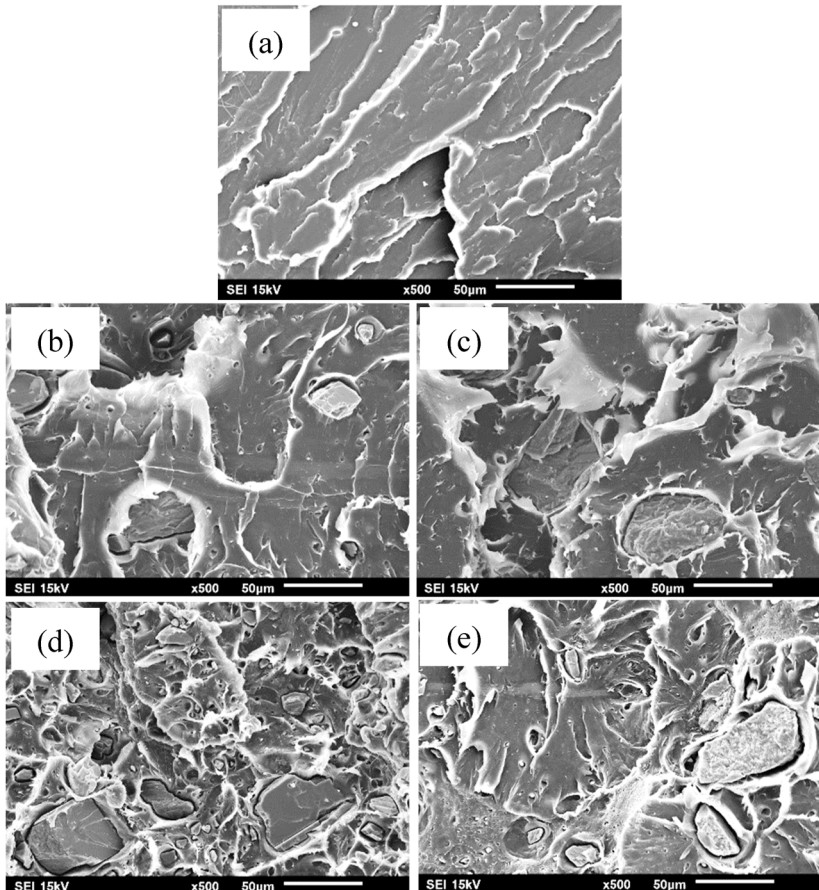

**Figure 8.** SEM images of the flexural fractured surfaces of PLA/CaCO$_3$ composites: (**a**) PLA, (**b**) PLA/ILS-5, (**c**) PLA/WE-5, (**d**) PLA/ILS-20, and (**e**) PLA/WE-20.

At higher filler concentrations (Figure 9d,e), debonding of the filler particles was more dominate especially for the white eggshell particles (Figure 9e) as denoted by the small holes in the matrix. In Figure 9d,e, the fractured surfaces have a flaky appearance and there is no specific direction of crack growth. The addition of higher filler loadings prevents plastic deformation from occurring, making the composite material more brittle. The industrial limestone composites (Figure 9d) appeared to have limited pull-out of particles from the matrix. The SEM results are consistent with the impact strength results of Figure 5, which show the PLA/ILS composites had improved impact strengths compared to the PLA/WE composites for both particles filler sizes.

### 3.1.5. Statistical Analysis

Statistical assessment using one-way analysis of variance (ANOVA) for tensile, flexure, and impact strengths for the composite materials are listed in Table 2. In all ANOVA tests, the F-critical was less than the F-value, which rejects the null hypothesis, indicating significant differences in the mean values of the mechanical strengths of the composites exist. ANOVA showed the means of the tensile, flexural, and impact strengths were statistically significantly affected by the type of filler (ILS and WE) and loading amounts added to the PLA matrix.

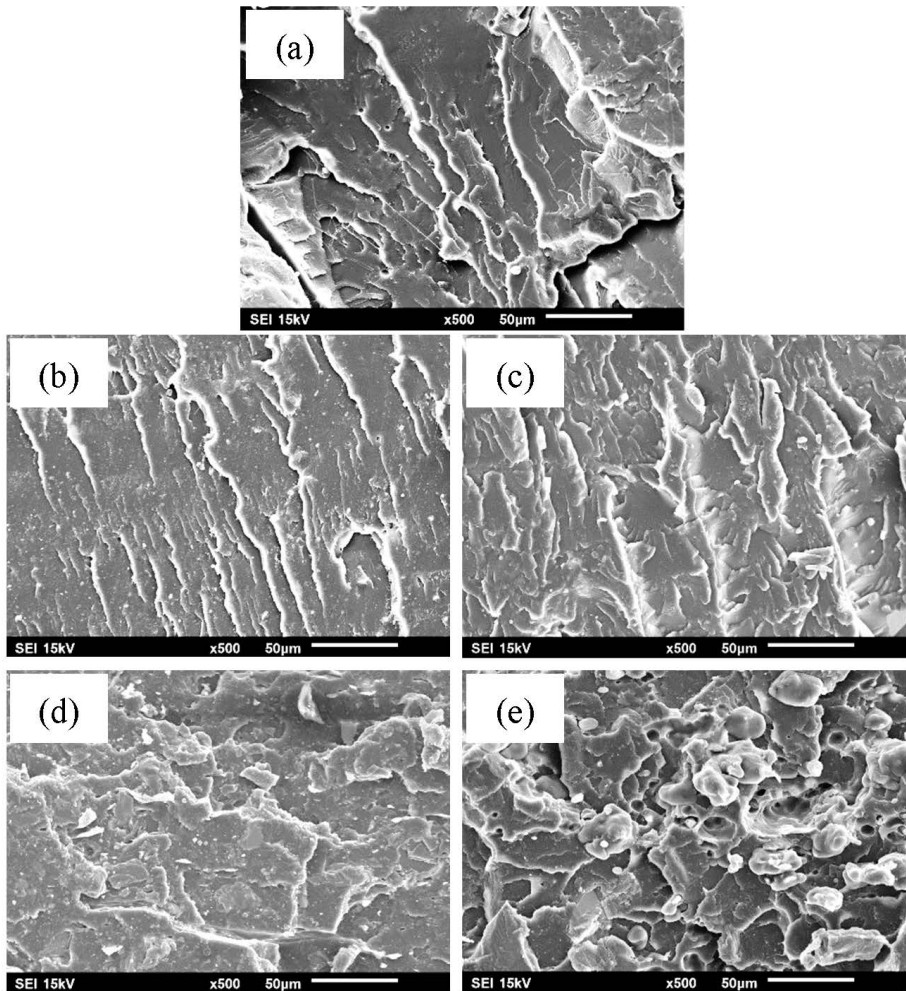

**Figure 9.** SEM images of the impact fractured surfaces of PLA/CaCO$_3$ composites: (**a**) PLA, (**b**) PLA/ILS-5, (**c**) PLA/WE-5, (**d**) PLA/ILS-20, and (**e**) PLA/WE-20.

**Table 2.** ANOVA results for tensile strength, tensile modulus, flexural strength, flexural modulus, and impact strength based on the calcium carbonate filler type (ILS and WE) in a PLA matrix.

| Source of Variation | SS | df | MS | F | F-Crit |
|---|---|---|---|---|---|
| Tensile strength (MPa) | | | | | |
| BG | 18,212 | 12 | 151.8 | 29.17 | 1.974 |
| WG | 234.2 | 45 | 5.205 | | |
| Tensile modulus (GPa) | | | | | |
| BG | 6.251 | 12 | 0.5209 | 13.71 | 1.952 |
| WG | 1.900 | 50 | 0.0380 | | |
| Flexural strength (MPa) | | | | | |
| BG | 24,631 | 12 | 2053 | 67.87 | 1.991 |
| WG | 1270 | 42 | 30.25 | | |
| Flexural modulus (GPa) | | | | | |
| BG | 1.798 | 12 | 0.1499 | 15.83 | 1.991 |
| WG | 0.3975 | 42 | 0.0095 | | |
| Impact strength (kJ·m$^{-2}$) | | | | | |
| BG | 1077 | 12 | 89.73 | 80.31 | 1.836 |
| WG | 130.7 | 117 | 1.117 | | |

BG, between groups; WG, within groups; SS, sum of squares; df, degree of freedom; and MS, mean square.

### 3.1.6. Water Absorption

The water absorption by pure PLA and PLA composites are shown in Figure 10 for two particle filler sizes: 63 μm (Figure 10a) and 32 μm (Figure 10b). Pure PLA samples absorbed the least amount of water and reached weight gain values of 0.895%, 0.910%, and 0.910% at 21, 28, and 35 days, respectively. In a related study, compression molded pure PLA supplied by Mitsui Chemicals Inc., Japan absorbed 1% of water after 30 days of immersion [41]. PLA has predominantly hydrophobic properties due to the methyl side group (e.g., $CH_3$), while its polar oxygen groups (e.g., carbonyl (C=O) groups, carboxyl (C-COO) end groups, and ether (C-O-C) linkages [42]) gives it moderate hydrophilic properties [43]. Therefore, although PLA is hydrophobic, it also has some hydrophilicity since PLA is known to be hydroscopic.

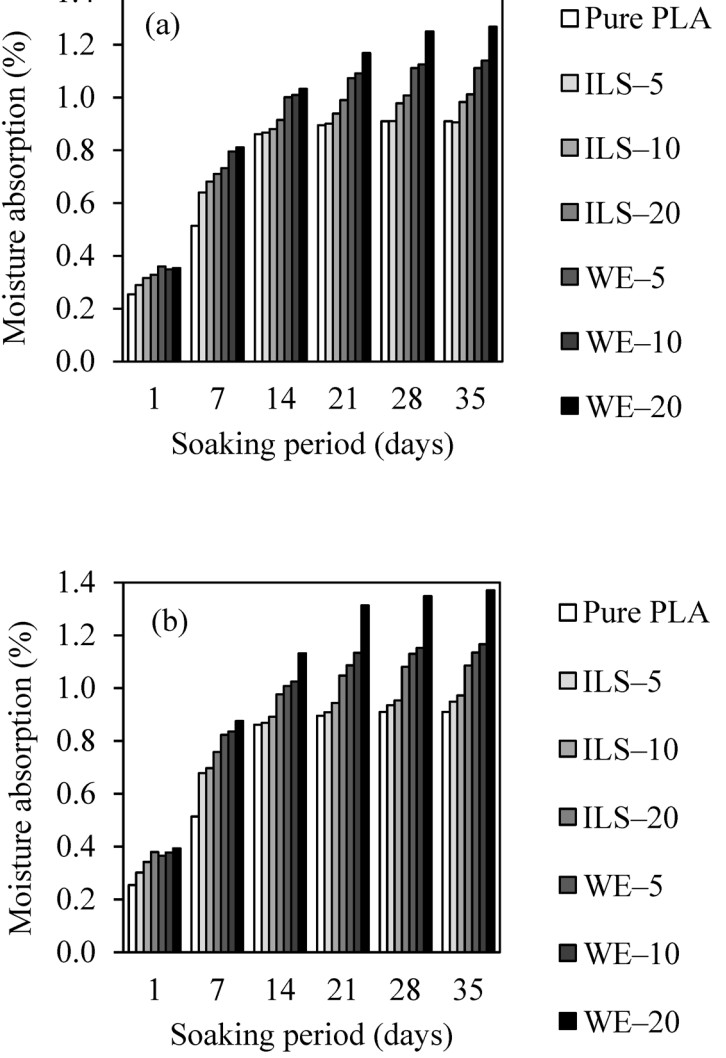

**Figure 10.** Weight gain due to moisture absorption of PLA and PLA composites for a total of 35 days in water: (**a**) 63 μm and (**b**) 32 μm particle size.

After 35 days of water immersion the PLA composite weight gains reached a maximum water uptake. For example, the composites containing 63 μm ILS and WE fillers in amounts of 5, 10, and 20 wt.%, had weight gains of 0.906%, 0.983%, and 1.01% and 1.11%, 1.14%, and 1.27%, respectively, while composites containing 32 μm fillers had weight gains of 0.935%, 0.954% and 1.08% and 1.13%, 1.15%, and 1.35%, respectively. The weight gains due to water absorption was observed to increase with increase in content of both filler types while the smaller particles had somewhat higher water weight gains. In an earlier study it

was found by X-ray diffraction (XRD) that the crystalline phase of both industrial limestone and white eggshells were based on calcium carbonate ($CaCO_3$) in the form of mineral calcite [39]. Pure calcite (100% $CaCO_3$) and limestone (95% $CaCO_3$ with 5% quartz ($SiO_2$)) were exposed to distilled de-ionized water and both were found to be very hydrophilic where limestone had slightly better wetting properties than calcite as given by the contact angle of $0°$ compared to $35°$, respectively. The difference was thought to be due to the surface roughness of the samples where limestone had a rough surface compared to the smooth surface of calcite [44]. In addition, the surface of limestone has hydrophilic/polar functional groups (hydroxyl (O-H) groups and carbonyl (C=O) groups), which enable easier intake of water [45]. Although the wetting behavior favors limestone, the composites containing limestone fillers in the current study absorbed less water than those containing eggshell fillers. The porous structures of both fillers could explain the differences in water absorption. From the literature, limestone porosity may differ from region to region, but has been reported to be in the range of 3% to 26% [46], while the surface porosity of the chicken eggshells has been reported to be between 1.07% to 4.43% [47]. The composites containing 32 μm fillers absorbed slightly more water than those containing 63 μm as shown in Figure 10. When particle size is reduced, a larger surface area enables more interaction with the liquid and increases the absorbable amount of water.

Overall, PLA composites containing 32 μm particles had improved mechanical properties compared to composites with 63 μm particles. Consequently, weight loss experiments together with leaching tests in distilled water for calcium contents and pH changes were carried out on the composites with the smaller particle sizes.

### 3.1.7. Weight Loss

As given in Figure 11, pure PLA had a minimal weight loss of about 0.106%, while composites had a slightly higher weight losses with white eggshell fillers showing more apparent reductions. For example, composites containing ILS and WE fillers in amounts of 5, 10 and 20 wt.%, had weight losses of 0.119%, 0.127%, and 0.129% and 0.165%, 0.179%, and 0.228%, respectively. The results of a similar study on water immersion of PLA (Ingeo 3051D) at 23 °C for 30 days had a weight reduction of about 0.07% [43]. It has been reported that small weight reductions can be attributed to marginal degradation of the PLA in water by hydrolysis at temperatures ranging from 20–45 °C [48,49]. In addition, hydrolysis and water absorption can proceed quicker when water temperatures are increased closer to and above the glass transition temperature ($T_g$) of the polymer [43]. Composite weight losses from long-term immersion in water may be due to the process of hydrolysis in a combined effect of PLA matrix degradation as well as susceptibility of hydrolysis between the organic matrix and inorganic industrial limestone/eggshells [50]. For instance, the biodegradable PLA 4043D used in the current study is an amorphous polymer with a low degree of crystallinity. The "D" indicates PLA has a poly(D-lactide) stereoisomer form with between 4.3–4.8% D-isomer contents [51]. An amount of 4–8% D-isomer in PLA was found to be an ideal range that does not allow crystallization to occur during the heating/cooling step [52]. For example, a previous study on PLA containing D-units in the poly lactide chain enhanced hydrolytic degradation over PLA without D-isomers [53]. Both weight and molecular weight losses were observed in hydrolysis of PLA in water [54]. A study reported small changes in molecular weight of a semi-crystalline PLA, (poly(L-Lactic acid)) (PLLA) that was injected molded and immersed in sea water for three months at 20 °C and 40 °C. The results showed a 14% and 48% reduction in molecular weights, respectively, which was attributed to the hydrolysis in the amorphous regions of the PLLA [55]. The phenomenon of hydrolysis proceeds faster in polymer amorphous regions than in crystalline regions since it is easier for water to permeate amorphous phases [52]. Degradation begins by uptake of water into the PLA due to the presence of polar and hydrophilic groups. The water molecules randomly react with ester linkages located along the PLA main chains, which produces chain scission and leads to molecular weight reduction of the PLA. The cleavage of ester bonds generates smaller chain fragments consisting of low molecular

weight lactic acid oligomers and monomers. These are soluble in water and can diffuse out of the polymer surface over time and reduce the properties of PLA [54]. Water in contact with the interfacial bonding between PLA and calcium carbonate particles can induce debonding followed by dislodging of the fillers close to the surface of the composites resulting in weight losses of the composite materials [56]. Calcium carbonate is stable in pure water and will not dissolve. It is anticipated that these solid particles will remain solid and get deposited at the bottom of the beakers. Therefore, additional weight losses when calcium carbonate filler contents are increased can be due to the additional water absorbed by the fillers. In Figure 11, composites containing eggshell fillers had a slightly greater weight loss than industrial limestone-filled composites possible due to their porosity and thus additional water uptake (Figure 10). A larger concentration of water may promote hydrolytic degradation in the PLA matrix as was highlighted in PLA/wood-flour [57] and in PLA/carbon nanotube (CNT) composites [48]. Based on the results in this current study, water absorption of PLA components submitted to a high moisture environment or submerged underwater for a long period of time should be considered.

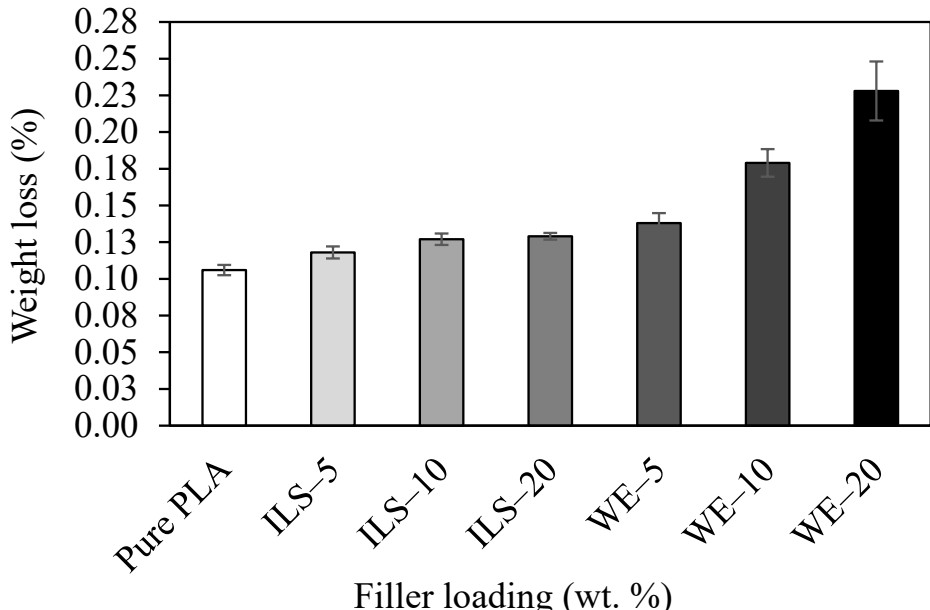

**Figure 11.** Weight loss over 35 days in distilled water for pure PLA and composite materials with different 32 μm particle sized filler loadings.

3.1.8. Leaching

To ascertain the extent of calcium carbonate leaving the PLA composites (32 μm filler particles), water samples from each beaker were analyzed for solid precipitates of calcium (Ca) concentrations using atomic absorption spectroscopy. The values are reported in milligrams (mg) of calcium as given in Figure 12. The beakers containing distilled water (as a control) and pure PLA samples measured 0 mg of Ca as was anticipated. The beakers containing the PLA/ILS composites had 0 mg of Ca content for all filler loadings, therefore the weight losses (Figure 11) are alleged to be due to PLA degradation (Figure 11). In contrast, the PLA/WE composites in filler loadings of 5, 10, and 20 wt.% had 0.34, 1.04, and 1.63 mg of Ca contents, respectively, in their distilled water. The weight loss of PLA/WE composites may be a contribution of both matrix and filler. The composite weight losses were attributed to both PLA and filler leaching, which infers higher filler loading, greater leaching, and more water absorption. To understand leaching between industrial limestone and eggshells in water, it is important to review the crystal structures of both minerals. In the literature, it is well known that eggshells have a calcite crystal structure identical to the mineral limestone form. The majority of these studies used X-ray diffraction to detect crystalline calcite, which cannot detect amorphous phases [58]. A recent study identified

calcite crystallites contained clusters of amorphous calcium carbonate using a transmission electron microscope (TEM) [59]. Generally, amorphous calcium carbonate is more soluble in water than crystalline calcium carbonate, which could explain the differences in leaching between ILS and WE PLA composites. In addition, industrial calcium carbonate solubility is well documented and is reported to have a small solubility in water (0.0013 g/100 mL at 25 °C) [60]; however, a review of the literature did not result in solubility experiments for eggshells and should be assessed.

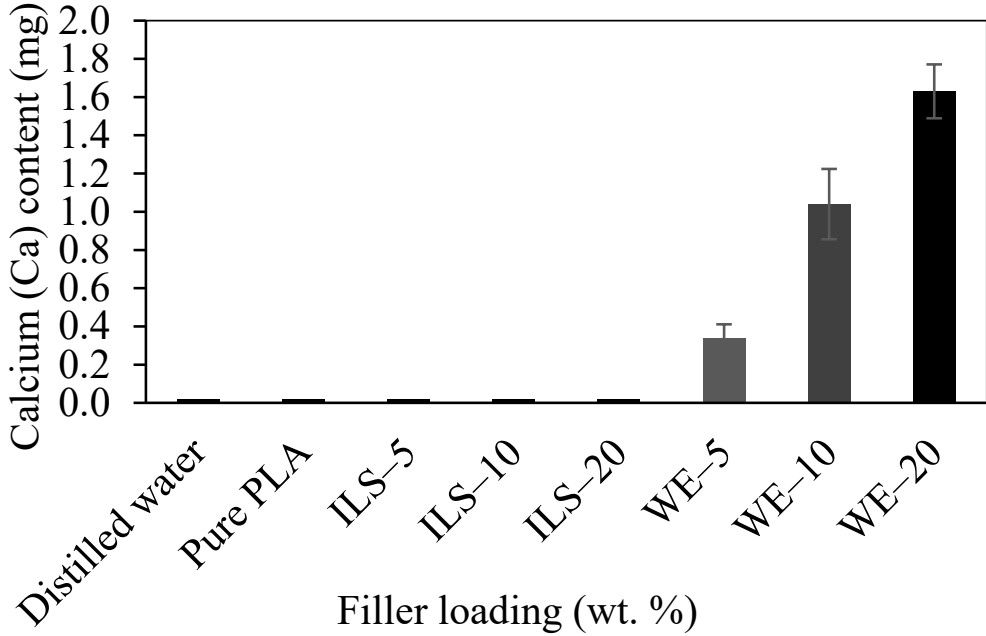

**Figure 12.** Calcium content after 35 days in distilled water for PLA composite materials with 32 μm particle sized filler loadings of 5, 10, and 20 wt.%.

3.1.9. pH Change of PLA Composites

After 35 days of water immersion, changes in pH values of the water medium in the beakers shown in Figure 13 could signify PLA matrix degradation and/or calcium carbonate fillers transferred from the composite materials to the water. PLA degradation in water is associated with a decrease in pH value due to release of lactic acid [61], while calcium carbonate-based fillers tend to increase the pH value due to its alkalinity [44]. The distilled water obtained from the dispenser in the laboratory had a pH of 6.20. After 35 days, the distilled water had a pH of 5.92 suggesting there may have been a slight absorption of carbon dioxide from the atmosphere. The pH of the distilled water after immersion of pure PLA samples had a pH of 5.83, which showed a slight reduction perhaps from a modest breakdown of PLA. The pH values for the PLA/ILS composites ranged from 5.85 to 5.90, which are similar to the unfilled composite. Obvious pH changes due to industrial limestone fillers were not observed possibly due to a lack of its extraction from the PLA as was also indicated by the Ca contents of these composites as given in Figure 12. The PLA containing the white eggshell fillers had pH values of 5.99, 6.09, and 6.48 for filler loadings of 5, 10, and 20 wt.%, respectively, indicating a slight increase compared to pure PLA. This upward trend in pH supports the calcium contents for these composites in Figure 12.

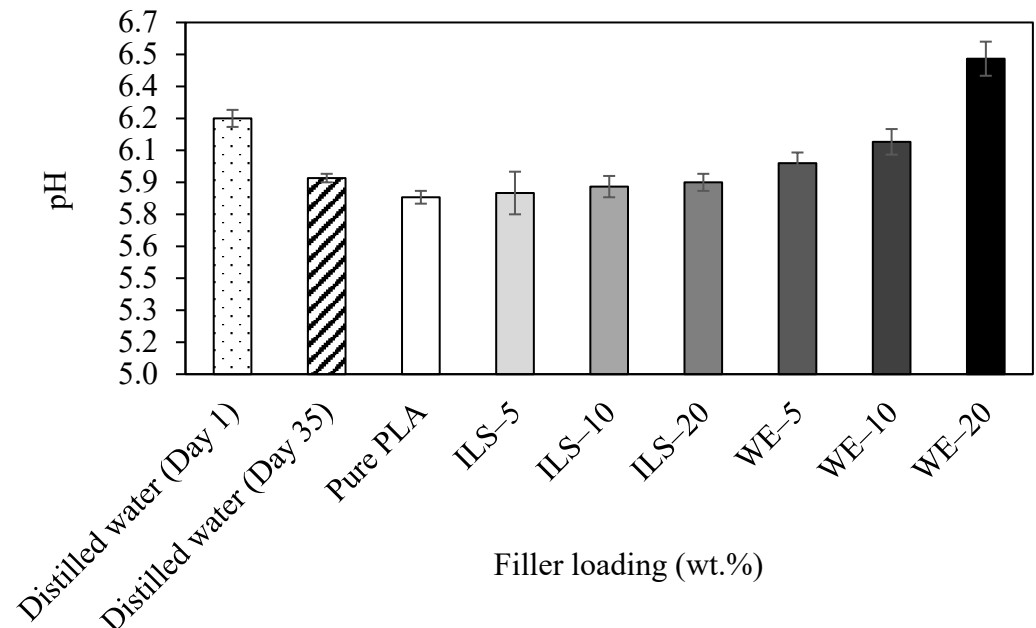

**Figure 13.** Change in pH of distilled water for PLA composites with 32 μm fillers after a 35-day absorption test.

## 4. Conclusions

This work compared industrial limestone and white eggshell filler powders in a PLA matrix. The tensile strength was found to decrease, while the tensile modulus improved with increased filler loading and both properties were controlled primarily by the amount of filler rather than the size of the particles. The flexural tests demonstrated PLA composites containing industrial limestone, smaller particles and 10 wt.% filler loadings were more suitable for members in bending. There was no toughening effect above that of pure PLA with the addition of either form of filler particles. SEM of the fractured surfaces depicted differences when the filler types and contents changed, which suggests the PLA matrix was modified to some level. ANOVA statistical assessment noted significant differences in the mean values of the mechanical strengths of the composites. Water absorption of PLA composites was higher than the pure PLA samples where composites containing white eggshell fillers absorbed more water than the industrial limestone, which could be due to the filler differences in their hydrophilic nature and porosities. As a result, waste white eggshells can successfully be used as fillers in PLA polymers to produce a completely bio-composite material.

**Author Contributions:** Conceptualization, D.C. and M.S.; methodology, D.C.; formal analysis, D.C.; investigation, D.C.; resources, D.C.; data curation, D.C.; writing—original draft preparation, D.C.; writing—review and editing, D.C. and M.S.; supervision, D.C.; project administration, D.C.; funding acquisition, D.C. All authors have read and agreed to the published version of the manuscript.

**Funding:** The authors would like to acknowledge the financial support of the Natural Sciences and Engineering Research Council of Canada (NSERC) under the Discovery Grant (RGPIN-2020-06701). The APC was funded by MDPI.

**Data Availability Statement:** The data presented in this study are available on request from the corresponding author. The data are not publicly available because the raw and processed data required to reproduce these findings cannot be shared at this time, as the data also form part of an ongoing study.

**Acknowledgments:** We would like to thank Ted Hudson of Burnbrae farms, Ontario, Canada for the in-kind donation of the waste eggshells.

**Conflicts of Interest:** The authors declare no conflict of interest. The funders had no role in the design of the study; in the collection, analyses, or interpretation of data; in the writing of the manuscript; or in the decision to publish the results.

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
