# Peer review of "Bio-Based White Eggshell as a Value-Added Filler in Poly(Lactic Acid) Composites"

_jcs, doi:10.3390/jcs7070278_

Round 1

Reviewer 1 Report

The paper shows some nice results on PLA composites with CaCO3 fillers, where limestone particles are compared to eggshell waste particles, and also the size of the particles is compared in terms of effects on PLA composite performance.

I have the following suggestions to improve the manuscript; please clarify the following:

Line 49-52: Could this be clarified? Do you mean that recycling of PLA does not need to produce pure lactic acid? Is this specific to the application where the recycled PLA is used in?

Line 61: Homogeneous distribution is a common challenge, though, so I wondered if they used a compatibilizer or modifier to enhance the distribution of the filler, avoid aggregation/agglomeration and improve compatibility with the polymer matrix?

Line 87-88: There are quite some studies, but of course not as many as those on conventional limestone powder

Line 237-242: Information should be provided on the amount of water in the beaker, and whether this changes over the 35 days. This will affect the concentration in then fluid, and thus the pH measurement

Line 266-267: The respective numbers are confusing. You mention three different amounts of fillers, but have 6 numbers in terms of impact on tensile modulus.

Line 269-272: The interpretation is not very convincing. Generally, agglomeration would be worse for higher contents of fillers, not for lower filler content.

Lines 309-314: The respective numbers are not clear. Divide into several sentences, and present them differently with only one variable changing “respectively”

Lines 315-316: This is not a clear interpretation. What are the factors that will change the distribution, and then what would you expect in terms of trends of distribution?

Lines 340-343: Again, the respective numbers are not clear. 

Line 347: Smaller particles would also have a higher energy and higher tendency to agglomerate

Lines 355-357: Again the respective numbers are not clear; change only one variable when reporting with “respectively”

Lines 436-439: I do not find the SEM images very clear or convincing. It would help if CaCO3 particles wer indicated or if SEM-EDS maps were added to show the evidence of the location of the CaCO3 particles.

A more elaborate interpretation on the higher uptake and leaching from white eggshells in comparison to industrial limestone may be needed. I would suggest looking into the structure of the materials. Also using SEM-EDS to stuyd the samples after leaching could give some more insights as well.

There are some mistakes in the grammar of the verb in some sentences, and a few on the English vocabulary in the text, but most is good

Reviewer 2 Report

The manuscript by Cree et al. studied the physical properties of PLA-based composites by using industrial limestone and white eggshells as fillers. They investigated the effects of filler concentration and filler size on mechanical properties, hygroscopicity and degradation of the composites. The manuscript is well-written and the data is sufficient, but some issues need to be addressed before considering the publication on JCS. 

1. The authors should clarify the innovation of this work. The filler content in this study is above 5 wt%, but previous studies have reported that the agglomeration will occur if the content is above 5 wt %, resulting in a decrease of the mechanical properties, and this is also evidenced by the results in this work. Why choices such a high filler content?

2. The word “weast eggshells” in line 3 of Abstract may be a typo.

3. What is the difference between 4043D PLA and pure PLA?

4. How to measure the sizes of these fillers? As shown in Figure 6, it can be found that the filler size is widely distributed and there is no significant size difference.

5. Scale bars are suggested to add in Fig. 1 and Fig. 2.

Round 2

Reviewer 1 Report

The authors have addressed the previous comments.

Reviewer 2 Report

The manuscript is acceptable in its current form.